# FT-Shield: A Watermark Against Unauthorized Fine-tuning in Text-to-Image Diffusion Models

## Abstract

Text-to-image generative models based on latent diffusion models (LDM) have demonstrated their outstanding ability in generating high-quality and high-resolution images according to language prompt. Based on these powerful latent diffusion models, various fine-tuning methods have been proposed to achieve the personalization of text-to-image diffusion models such as artistic style adaptation and human face transfer. However, the unauthorized usage of data for model personalization has emerged as a prevalent concern in relation to copyright violations. For example, a malicious user may use the fine-tuning technique to generate images which mimic the style of a painter without his/her permission. In light of this concern, we have proposed FT-Shield, a watermarking approach specifically designed for the fine-tuning of text-to-image diffusion models to aid in detecting instances of infringement. We develop a novel algorithm for the generation of the watermark to ensure that the watermark on the training images can be quickly and accurately transferred to the generated images of text-to-image diffusion models. A watermark will be detected on an image by a binary watermark detector if the image is generated by a model that has been fine-tuned using the protected watermarked images. Comprehensive experiments were conducted to validate the effectiveness of FT-Shield.

## 1 Introduction

Generative models, particularly Generative Diffusion Models (GDMs) (Ho et al., 2020; Song et al., 2020b; Ho & Salimans, 2022; Song et al., 2020a), have witnessed significant progress in generating high-quality images from random noise. Lately, text-to-image generative models leveraging latent diffusion (Rombach et al., 2022) have showcased remarkable proficiency in producing specific, detailed images from human language descriptions. Based on this advancement, fine-tuning techniques such as DreamBooth (Ruiz et al., 2023) and Textual Inversion (Gal et al., 2022) have been developed. These methods enable the personalization of text-to-image diffusion models, allowing them to adapt to distinct artistic styles or specific subjects. For instance, with a few paintings from an artist, a model can be fine-tuned to adapt to the artistic style of the artist and create paintings which mimic the style. However, the proliferation of these methods has sparked significant copyright concerns. There are particular concerns about the potential misuse of these techniques for unauthorized style imitation or the creation of deceptive human facial images. Such actions infringe on the rights of original creators and breach the sanctity of intellectual property and privacy.

To protect data from being unauthorizedly used for text-to-image model fine-tuning, there is a line of research (Van Le et al., 2023; Liang et al., 2023; Shan et al., 2023; Salman et al., 2023) focusing on designing perturbations in the data to prevent any model learning from the data. In these works, *adversarial techniques* are applied to make the images resistant to subject-driven synthesis techniques, causing the text-to-image model to learn subjects that are substantially divergent from the original intentions. However, one issue is that these protective methods can inadvertently disrupt authorized uses of the safeguarded images. In reality, we may still allow some people to learn the image (such as for academic research purpose). To tackle this challenge, another line of research (Ma et al., 2023; Wang et al., 2023) considering *watermarking techniques*. Unlike the previous methods, watermarking does not disrupt the regular fine-tuning process of the models. It allows the intellectual property to be used for proper reasons, while also acting as a warning and a way to collect proof

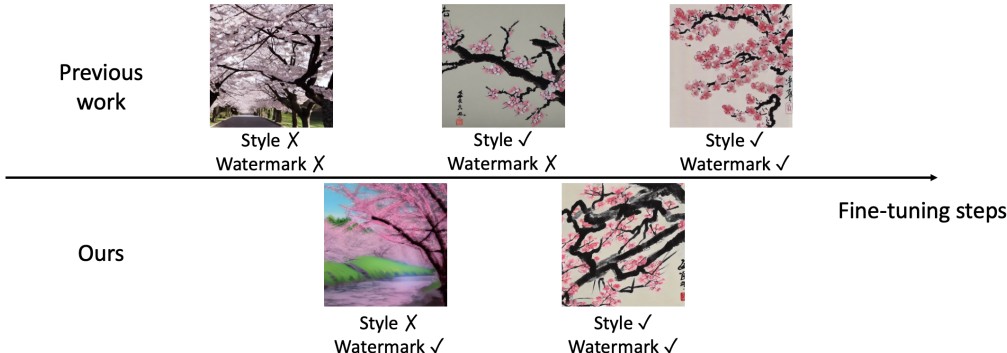

Figure 1: Generated images from fine-funed text-to-image models: a comparison of watermarking methods at different fine-tuning steps.

against improper uses. Specifically, watermarking techniques work via embedding a covert signal or pattern into the original data which is imperceptible to human eyes but can be detected by a specialized watermark detector. When the watermarked images are used for fine-tuning text-to-image models, the resulting generated images are expected to inherit this watermark, acting as an indelible signature. By employing a watermark detector, one can identify the presence of these watermarks in the generated content. Thus, if an entity produces images using the protected data, the watermark will serve as an evidence of infringement.

There have been existing watermarking methods (Ma et al., 2023; Wang et al., 2023) designed for protecting the copyright of images against text-to-image model fine-tuning. However, they often require sufficient fine-tuning steps to ensure that the watermark on the images can be adequately assimilated by the generative model and subsequently, transferred to the generated images. In real practice, even when the fine-tuning steps are not extensive enough for the watermarks to be learned, the style of the images can already be acquired by the model. As shown in Figure 1, as the fine-tuning steps increase, the style of the paintings can be learned by the model earlier than the watermark by Ma et al. (2023). Therefore, the watermarks may fail to act as an effective protection because the offender can evade the watermark by simply reducing the fine-tuning steps. One key reason for this issue is that those watermark generating procedures predominantly adhere to traditional watermarking strategies which are intended to trace the source of an image rather than protect the copyright against diffusion models. Therefore, integral features of their watermarks may not be promptly and precisely assimilated by the diffusion models.

In this work, we propose a novel watermarking framework, **FT-Shield**, tailored for data's copyright protection against the **F**ine-**T**uning of text-to-image diffusion models. In particular, we introduce a training objective which incorporates the training loss of the diffusion model for the generation of the watermark. By minimizing this training objective, we ensure that the watermark can be quickly learned by the diffusion model in the very early stage of fine-tuning. As shown in Figure 1, by using our method, even when the style has not been learned by the diffusion model, the watermark can already be learned by the diffusion model and detected on the generated images. The effectiveness of the proposed method is verified through experiments across various fine-tuning techniques including DreamBooth, Textual inversion, Text-To-Image Fine-tuning (von Platen et al., 2022) and LoRA (Hu et al., 2021), applied to both style transfer and object transfer tasks across multiple datasets.

## 2 RELATED WORK

### 2.1 TEXT-TO-IMAGE DIFFUSION MODEL AND THEIR FINE-TUNING METHODS

Diffusion models (Ho et al., 2020; Song et al., 2020b; Ho & Salimans, 2022; Song et al., 2020a) have recently achieved remarkable advancements in the realm of image synthesis, notably after the introduction of the Denoising Diffusion Probabilistic Model (DDPM) by (Ho et al., 2020). Building upon the DDPM framework, Rombach et al. (2022) presented the Latent Diffusion Model (LDM). Unlike conventional models, LDM conducts the diffusion process within a latent space derived from a pre-trained autoencoder, and generates hidden vectors by diffusion process instead of directly generating the image in pixel space. This strategy enables the diffusion model to leverage the robust semantic features and visual patterns imbibed by the encoder. Consequently, LDM has set new standards in both high-resolution image synthesis and text-to-image generation.

Based on the advancement in the text-to-image diffusion model, multiple fine-tuning techniques (Gal et al., 2022; Ruiz et al., 2023; Hu et al., 2021) have been developed. These methods enable the personalization of text-to-image diffusion models, allowing them to adapt to distinct artistic styles or specific subjects. Specifically, DreamBooth (Gal et al., 2022) works by fine-tuning the denoising network of the diffusion model to make the model associate a less frequently used word-embedding with a specific subject. Textual Inversion (Ruiz et al., 2023) tries to add a new token which is bound with the new concept to the text-embedding of the model. And LoRA (Hu et al., 2021) proposed to add pairs of rank-decomposition matrices to the existing weights of the denoise network and only train the newly added weights in the fine-tuning process.

## 2.2 IMAGE PROTECTION METHODS

In literature, there are two main types of protection methods for intellectual property against text-to-image model fine-tuning: (1) designing perturbations in the data to prevent any model learning from the data. (2) designing watermarks so that protectors can detect whether generated images infringe the intellectual property or not.

**Adversarial methods.** Adversarial methods protect the data copyright by taking the advantage of the idea in evasion attacks. They treat the unauthorized generative models as the target model of an attack, and the attack tries to reduce generative models' ability in learning from images. Thus, the protected images are designed as adversarial examples which reduce the unauthorized models' performance and further protect themselves. GLAZE is first proposed to attack the extracted feature by the encoder part in Stable Diffusion and prevent the style of images from being learned by others (Shan et al., 2023). Van Le et al. (2023) proposed Anti-DreamBooth to prevent a method of fine-tuning Stable Diffusion, DreamBooth (Ruiz et al., 2022). Liang et al. (2023) use a Monte Carlo method to generate adversarial examples to evade the infringement from Textual Inversion (Gal et al., 2022). (Salman et al., 2023) propose to modify the pictures to protect them from the image editing applications by Stable Diffusion in case the pictures are used to generate any images with illegal or abnormal scenarios. Although these methods can protect the data copyright, it destroys the authorized uses for certain generation purpose. Thus, tracing the usage and identifying the unauthorized infringement are important.

**Watermarking methods.** Watermarking techniques have also been considered to protect the intellectual property of images against text-to-image model fine-tuning. Wang et al. (2023) proposed to apply an existing backdoor method (Nguyen & Tran, 2021) to embed unique signatures into the protected images. This aims to inject extra memorization into the text-to-image diffusion models fine-tuned on the protected dataset so that unauthorized data usage can be detected by checking whether the extra memorization exists in the suspected model. However, a limitation of their approach is the assumption that the suspicious diffusion model is readily accessible to the data protector. This might not be realistic, as malicious entities might only disclose a handful of generated images but hide the fine-tuned model. Another work by Ma et al. (2023) introduces a method that trains a watermark generator and detector simultaneously. The detector is then further fine-tuned using the images generated by the fine-tuned model. However, as discussed in Section 1, the issue of this technique is that it provides no guarantee that their watermark can be learned by the model earlier than the image's style. The shortcomings of current methods have highlighted the need for a more robust and effective watermarking scheme.

## 3 METHOD

In this section, we first define the problem and introduce the necessary notations. Then we present the details of the proposed FT-Shield by introducing the watermark generation procedure and the training method for the watermark detector.

## 3.1 PROBLEM FORMULATION

In the scenario of copyright infringement and protection considered in this work, there are two roles: (1) a **data protector** that possesses the data copyright, utilizes watermarking techniques before the data are released, and tries to detect if a suspected model is fine-tuned on the protected images, and (2) a **data offender** that utilizes the protected data for text-to-image diffusion model fine-tuning without the permission from the data protector. The data offenders have complete control over the fine-tuning and sampling processes of the text-to-image diffusion models, while the data protectors

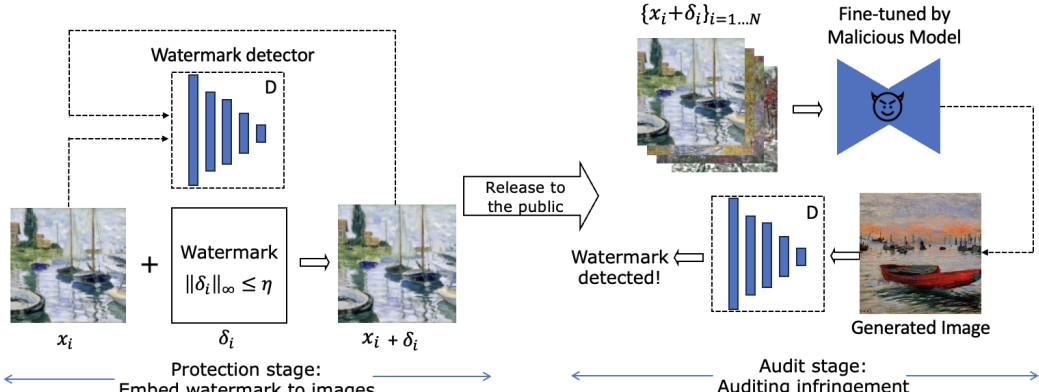

Figure 2: An overview of the two-stage watermarking protection process

can only modify the data they own before the data are released and access images generated by the suspected model.

As shown in Figure 2, the protection process consists of two stages: the protection stage and the audit stage. In the **protection stage**, the data protector protects the images by adding imperceptable watermarks to the images. Specifically, given that the size of the protected dataset is $N$, the target is to generate sample-wise watermark $\delta_i$ for each protected image $x_i, \forall i = 1...N$. Then these watermarks are embedded into the protected images $\hat{x}_i = x_i + \delta_i$. Correspondingly, the data protector will train a binary watermark detector, $D_w$, to test whether there is a watermark on the suspect image. To ensure that the watermarks will not lead to severe influence of the image quality, we limit the budget of the watermark by constraining its $l_\infty$ norm ($\|\delta_i\|_\infty \leq \eta$) to control the pixel-wise difference between the two images $x_i$ and $\hat{x}_i$. In the subsequent **audit stage**, if the protectors encounters suspected images potentially produced through unauthorized text-to-image models fine-tuning, they utilize the watermark detector $D_\omega$ to ascertain whether these images have infringed upon their data rights.

## 3.2 WATERMARK GENERATION

As discussed in Section 1, the challenge of building a robust and reliable watermark framework is to ensure that the watermark can be quickly and precisely integrated to the diffusion models. This integration should occur as rapidly as the model learns the style during the incremental fine-tuning steps. In essence, the pivotal aim is to generate specific features that are highly sensitive to the diffusion model, ensuring the model prioritizes learning the watermark features along with or before the style of the images. To achieve this goal, given $n$ samples to be protected, we construct a training objective for the watermark as follows:

$$\min_{\{\delta_i\}_{i \in [n]}} \min_{\theta_1} \sum_{i \in [n]} L_{dm}(\theta_1, \theta_2, x_i + \delta_i, c) \text{ s.t. } \|\delta_i\|_\infty \leq \eta \tag{1}$$

where $\theta_1$ represents the parameters of the UNet (Ronneberger et al., 2015), which is the denoise model within the text-to-image model structure, $\theta_2$ denotes the parameters of the other part of the diffusion model, and $c$ is the prompt for the image generation. The function $L_{dm}$ denotes the fine-tuning loss of the text-to-image diffusion model:

$$L_{dm}(\theta_1, \theta_2, x_0, c) = \mathbb{E}_{x_0, t, c, \epsilon \in N(0, I_d)} \|\epsilon - \epsilon_{\theta_1, \theta_2}(x_t, t, c)\|_2, \tag{2}$$

with $d$ as the dimension of the training images in the latent space, $x_0$ as the input image, $t$ as the timestep and $x_t$ as the input image with $t$ steps' noise in the diffusion process. The above training objective aims to figure out the best perturbation $\delta_i$ for each sample $x_i$ so that the loss of a diffusion model trained on these perturbed samples can be minimized. In other words, we want to ensure that with the existence of the optimized watermark, the training loss of the diffusion model can be reduced rapidly in its training process. Intuitively, this process engenders a "shortcut" feature within the watermark which can be quickly learned and emphasized by the diffusion model. By ensuring that the model rapidly assimilates this shortcut feature, we enhance the probability that the watermark can be successfully transferred to images generated by the model fine-tuned on the watermarked datasets, even when the fine-tuning does not involve many steps. We solve this bi-level optimization problem by alternatively updating the perturbation and the parameters of the diffusion model. Details about the algorithm are provided in Appendix A.

**The Design of Fine-tuning Loss and Prompt.** There have been multiple fine-tuning methods for text-to-image diffusion model and each of them involves different formulations of fine-tuning loss and different parts of the model to be updated. Therefore, it is crucial to ensure that our watermarking technique remains effective across different implementations of text-to-image model fine-tuning. This requires us to carefully select the specific $L_{dm}$ in Equation 1 and the caption $c$ used in watermark training. In terms of the particular $L_{dm}$ considered in Equation 1, to maintain simplicity and coherence, we focused on the most fundamental Text-to-Image Fine-tuning Method (von Platen et al., 2022). This method aligns with the original training objectives of the text-to-image model and involves updates only to the UNet. The experiments in Section 4.2 demonstrate that the watermarks generated with this formulation are able to be assimilated well by different fine-tuning methods, even those do not modify the UNet of the model such as Textual Inversion (Ruiz et al., 2023). For the caption $c$, we employ a simplistic and consistent caption format for every image within the dataset in the watermark generation process. This consistency ensures robustness in varying conditions. Specifically, for the images associated with style transfer tasks, the caption is "*A painting by \**", with \* denoting the artist's name. For images used for object transfer, each is labeled by "*a photo of \**", where \* indicates the object's category, such as 'dog', 'toy', or 'person'.

### 3.3 WATERMARK DETECTOR

After the watermarks are generated, we require a binary watermark detector to detect if an image generated by the suspected model contains a watermark or not. However, considering that the size of the protected datasets is small, directly using the watermarked images and clean images to train the detector would result in poor performance due to overfitting. To tackle this challenge, we propose to use generated images from the fine-tuned text-to-image models to augment the dataset for training the detector. In the following, we first introduce the training procedure of the detector and then give the details about how to augment the training data of the detector.

**Detector Training.** We transform the detection of watermark into a binary classification task distinguishing clean and watermarked image. The training objective of the detector, i.e. the binary classifier $D_\omega$, is formulated as:

$$L_{cl}(\omega) = \sum_{i \in [n]} \left[ -\log(1 - D_\omega(x_i)) - \log D_\omega(x_i + \delta_i) \right], \tag{3}$$

which is the cross-entropy loss of the classification for determining whether an images is embedded with a watermark or not, and $\omega$ refers to the parameters of $D$. To save memory and computation cost, we consider applying the classification directly to the latent features of each image. In other words, we need to apply a VAE within the latent diffusion model's structure to encode each image into its latent space before they are used to train the classifier.

**Augmentation with Generated Images.** Given that the size of the protected dataset can be very small, the watermark detector trained on only the protected dataset is unreliable. To address this issue, we propose using generated images from the fine-tuned text-to-image models to augment the training set of the detector. This process unfolds as follows: we first fine-tune the text-to-image diffusion model using a particular fine-tuning method with both clean and watermarked datasets, yielding two separately fine-tuned models. Subsequently, these models are employed to generate two distinct sets of data. Data generated from the model fine-tuned using clean data are incorporated to the original clean dataset, and the images generated from the model fine-tuned with watermarked images are utilized to augment the watermarked dataset for the training of the detector.

If the data protector has the knowledge about the specific fine-tuning method the offender uses, he/she can directly use the images generated from that specific fine-tuning method to augment the data for the classifier training. In the case that the data protector has no knowledge on the fine-tuning method, we have experiments in 4 to demonstrate that with a relatively higher watermark budget, the detector for one fine-tuning method can exhibit a decent adaptability when applied to the data generated by an alternative fine-tuning method. This transferability is notably prevalent in the classifier trained with the augmentation of the data generated by LoRA.

## 4 EXPERIMENT

In this section, we evaluate the efficacy of our FT-Shield across various fine-tuning methods, subject transfer tasks and different datasets. We first introduce our experimental setups in Section 4.1. In Section 4.2, we evaluate and analyze our approach in terms of its detection accuracy and influence

Table 1: Detection accuracy of watermark

|  |  | ours ($\eta = 4/255$) | | ours ($\eta = 2/255$) | | GW | | IM | |
| --- | --- | --- | --- | --- | --- | --- | --- | --- | --- |
|  |  | TPR↑ | FPR↓ | TPR↑ | FPR↓ | TPR↑ | FPR↓ | TPR↑ | FPR↓ |
| Style | DreamBooth | **99.50%** | **0.18%** | 98.68% | 0.87% | 93.31% | 3.81% | 84.27% | 4.18% |
|  | Textual Inversion | **96.12%** | 3.03% | 93.55% | 5.25% | 78.75% | 12.70% | 68.07% | **0.25%** |
|  | Text-to-image | **98.77%** | **1.28%** | 96.77% | 3.54% | 75.41% | 30.03% | 71.87% | 5.30% |
|  | LoRA | **97.65%** | **2.67%** | 93.37% | 6.17% | 67.28% | 22.72% | 67.97% | 9.78% |
| Object | DreamBooth | **98.93%** | 1.23% | 97.60% | **1.13%** | 91.39% | 3.50% | 75.97% | 1.20% |
|  | Textual Inversion | **97.73%** | **1.67%** | 97.23% | 1.97% | 88.22% | 3.95% | 58.41% | 22.20% |

("↑" means a higher value is better. "↓" means a lower value is better.)

on the image quality during both the protection and audit stages. Then we further investigate our approach from Section 4.3 to Section 4.5 in terms of its performance under fewer fine-tuning steps, transferability across different fine-tuning methods and robustness against image corruptions. And we have demonstrated the ablations studies in 4.6.

## 4.1 EXPERIMENT SETTINGS

**Model, Task, Dataset and Baselines.** We conducted our experiments using the Stable Diffusion as the pretrained text-to-image model. The size of all the images is 512x512. The primary focus was on two tasks: style transfer and object transfer. For style transfer, we utilized 10 datasets sourced from WikiArt, each containing between 20 to 40 images. The object transfer task was more diversified, incorporating datasets of two lifeless objects, and three individual human faces, with each dataset containing five images. We adopted different fine-tuning methods for different tasks: for style transfer, the methods included DreamBooth (Ruiz et al., 2023), Textual-inversion (Gal et al., 2022), Text-to-Image Fine-tuning (von Platen et al., 2022), and LoRA (Hu et al., 2021), while for object transfer, only DreamBooth and Textual-inversion were utilized because the performance of the other two methods is not good. We considered the watermarking methods Inject Memorization (IM) (Wang et al., 2023) and Gen-Watermark (GM) (Ma et al., 2023) which are also proposed for copyright protection against the fine-tuning of text-to-image diffusion model as our baselines.

**Implementation Details.** For watermark generation, we consider watermark budgets of 4/255 and 2/255 for each dataset. The watermarks are trained with 5-step PGD (Madry et al., 2018) with step size to be 1/10 of the $l_\infty$ budget. For watermark detection, we adopted ResNet18 (He et al., 2016) with the Adam optimizer, setting the learning rate to 0.001 and the weight decay to 0.01. In detection stage, we use 60 and 30 prompts for image generations in style transfer and object transfer tasks, respectively. Details about the prompts and hyperparameters of the fine-tuning methods are in Appendix C, and B.

**Evaluation Metrics.** We evaluate the effectiveness of our watermarking approach from two perspectives: 1) its influence on the quality of the released protected images and the generated images; 2) its detection accuracy on the generated images of fine-tuned text-to-image models.

- *Image Quality.* We use FID (Heusel et al., 2017) to assess the impact of watermarks on image quality. Specifically, we measure the visual discrepancies between the original and watermarked images to evaluate its influence on the released images' quality. And we also calculate the FID between the images generated from models fine-tuned on clean images with those generated from models fine-tuned on watermarked images to measure the watermark's influence on the generated images. A lower FID indicates better watermark invisibility.

- *Detection Accuracy.* The efficacy of the watermark detection on the generated images is assessed by two metrics. First, the detection rate (or the true positive rate, TPR) quantifies the proportion of instances where the detector accurately identifies images produced by models fine-tuned on watermark-protected images. Second, the false positive rate (FPR) represents the percentage of instances where the detector mistakenly flags images without watermarks as watermarked.

## 4.2 MAIN RESULTS

In this section, we show that our FT-Shield can achieve good performance in two aspects: 1) a high detection accuracy; and 2) a small influence on the quality of images. In experiments, we consider the scenario that the specific fine-tuning method of the suspected model is known by the data protector and the protector can use the generated images of that method as augmentation to train the watermark detector. Note that in Section 4.4, we will demonstrate the transferability o our watermark detector that can ensure its effectiveness when the fine-tuning method is unknown

Table 2: FID ↓ between clean and watermark images in released and generated images

| | | | FT-Shield ($\eta : 4/255$) | FT-Shield ($\eta : 2/255$) | GW | IM |
|---|---|---|---|---|---|---|
| Style | Released | | 20.80 | **6.79** | 58.04 | 65.50 |
| | Generated | DreamBooth | 62.25 | 46.96 | **46.42** | 49.77 |
| | | Textual Inversion | 67.99 | 59.25 | **41.99** | 62.73 |
| | | Text-to-image | 33.66 | **33.00** | 38.40 | 35.71 |
| | | LoRA | 32.76 | 29.99 | 30.12 | 33.30 |
| Object | Released | | 29.45 | **10.01** | 46.25 | 57.86 |
| | Generated | DreamBooth | 49.19 | 41.93 | 37.57 | **37.21** |
| | | Textual Inversion | 92.87 | **62.67** | 102.32 | 79.58 |

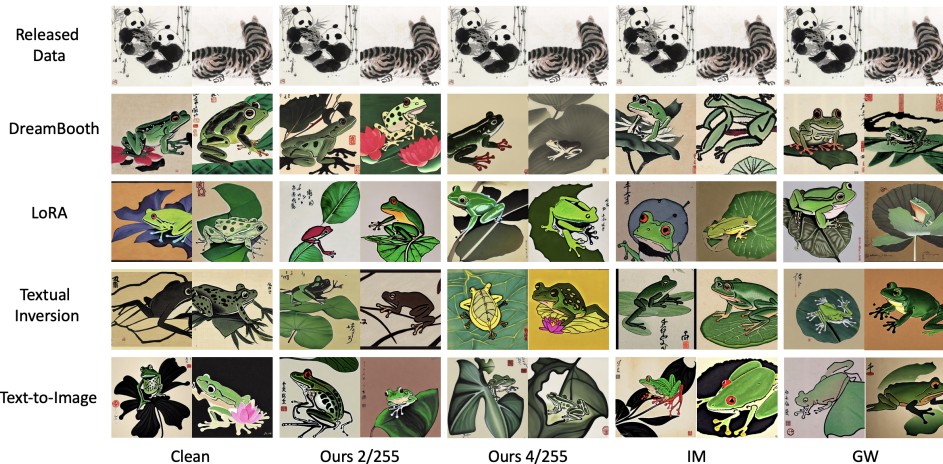

Figure 3: Examples of watermarked images (first line) and generated images (other lines) in the style of artist Beihong Xu. The prompt of generation: A frog on a lotus Leaf in the style of [V]

to the data protector. In the evaluation of detection accuracy, we measure the TPR and FPR of the watermark detector in the generated images. And we use FID to evaluate the influence of the watermark in the released and generated images. The average of the main metrics across multiple datasets for the two transfer tasks are demonstrated in Table 1 and Table 2, respectively. More statistics about the TPR and FPR are shown in Figure 4. Examples of the watermarked released images and generated images can be found in Figure 3.

According to the results in Table 1 and Table 2, our method is able to protect the images with the highest TPR and lowest FID values in released dataset among all the fine-tuning methods in both style transfer and object transfer tasks. With an $l_\infty$ budget to be 4/255, the TPR of the watermark detection can achieve nearly 100% across all the fine-tuning methods. At the same time, the FPR is also very close to 0. Even constrained by a small budget with an $l_\infty$ norm to be 2/255, FT-Shield can still achieve a TPR consistently higher than 90% and an FPR no higher than 7%. As shown in Figure 4, in general, the variance of the TPR and FPR of our FT-Shield are consistently smaller than the baseline methods, indicating a more stable ability in protecting the images. Although in some cases the FID of the generated images of DreamBooth and Textual Inversion is relatively higher than the baseline methods, our method consistently demonstrates a much higher watermark detection accuracy. This ensures the success of the detection of unauthorized use. In comparison, baseline methods Gen-Watermark (Ma et al., 2023) and IM (Wang et al., 2023) has a decent TPR and FPR in DreamBooth, but both of them cannot achieve a good accuracy in the images generated by other fine-tuning methods. This indicates that they are not able to provide reliable protection against those fine-tuning methods.

## 4.3 PERFORMANCE UNDER INSUFFICIENT FINE-TUNING STEPS

In this subsection, we provide more evidence that the watermarks generated by our FT-Shield can be better assimilated by the fine-tuning process of diffusion models. Based on Dream-Booth, we conduct model fine-tuning with much fewer steps compared with standard experiments. And we apply our watermark detector generated in the standard experiment to calculate

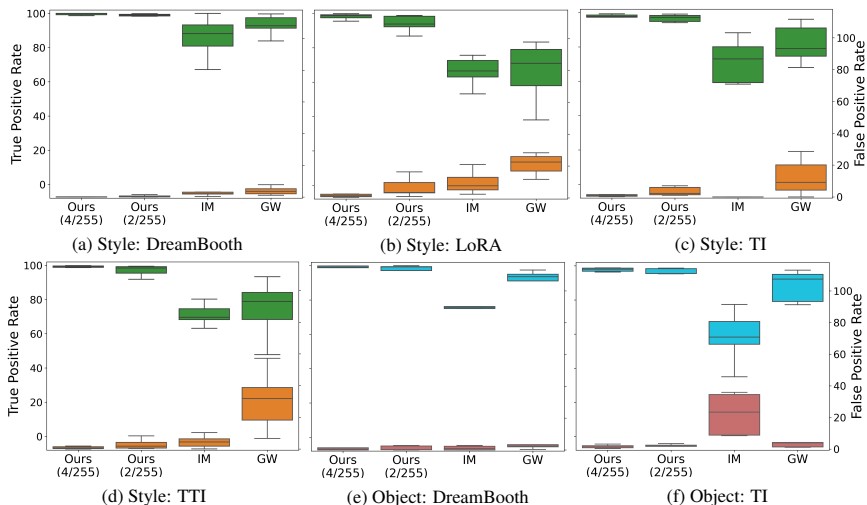

Figure 4: TPR (green/blue boxes) and FPR (orange/red boxes) of watermark detector

the detection rate (TPR) of the watermark. These experiments are mainly done with the style transfer tasks using the paintings of artist Claude Monet. The results are demonstrated in Table 3. As shown in Table 3, FT-Shield consistently achieves the highest detection rate of the watermark when the fine-tuning steps are insufficient. Even when the fine-tuning steps are as few as 10, the detection rate of watermark can achieve 56%. With only 300 steps, the watermark detection can achieve nearly 100%. In comparison, the two baseline methods require much more steps to achieve a high detection rate. This indicates that our FT-Shield can be assimilated by the text-to-image model much earlier than the two baseline methods.

Table 3: Detection Rate (TPR) under fewer fine-tuning steps

| Steps | Ours (4/255) | GW | IM |
|---|---|---|---|
| 10 | 56.17% | 32.67% | 3.50% |
| 20 | 66.00% | 33.50% | 7.33% |
| 50 | 65.96% | 52.67% | 2.00% |
| 100 | 66.94% | 41.00% | 1.83% |
| 200 | 76.50% | 57.97% | 13.33% |
| 300 | 97.17% | 66.67% | 54.67% |
| 900 | 100.00% | 99.72% | 93.83% |

### 4.4 TRANSFERABILITY ACROSS DIFFERENT FINE-TUNING METHODS

In this subsection, we demonstrate that the detector trained with the images generated by one fine-tuning method can achieve good adaptability to the data generated by another fine-tuning method. In Table 4, we evaluate the detection accuracy (detector's average accuracy in the watermark class and clean class) of the classifier which is trained with the images from one fine-tuning method on the images generated by the other three fine-tuning methods. The experiments are mainly based on

Table 4: Transferability of the watermark detectors

| FT Method that the classifier trained on | Ours (4/255) | Ours (2/255) | GW |
|---|---|---|---|
| DreamBooth | 78.51% | 75.42% | 64.60% |
| Textual Inversion | 84.44% | 62.10% | 56.52% |
| Text-to-Image | 88.32% | 78.48% | 68.27% |
| LoRA | 86.42% | 82.83% | 65.29% |
| Avg. | 84.42% | 74.71% | 63.67% |

the style transfer task and the performance is compared with Gen-Watermark (Ma et al., 2023), which also requires the generated images in developing the watermark detector. According to the results demonstrated in Table 4, our detectors consistently transfer well to other fine-tuning methods, achieving average accuracy of about 85% and 75% for budgets 4/255 and 2/255, respectively. Notably, the classifier trained with the images from LoRA achieves a transfer accuracy higher than 80% for both budgets. In comparison, the classifiers from GW exhibit inferior adaptability. Each of their classifiers achieves a transfer accuracy lower than 70%.

### 4.5 ROBUSTNESS OF WATERMARK

The robustness of a watermark refers to its ability to remain recognizable after undergoing various modifications, distortions, or attacks. It is a crucial property of watermark because during the images' circulation, the watermarks may be distorted by some disturbances, such as JPEG compression. The data offender may also use some methods to remove the watermark. In this subsection, we show that our watermarking method can be robust against multiple types of corruptions when proper augmentation is considered in the training of the watermark detector. In the experiment in Table 5, we consider four types of image corruptions including JPEG compression, Gaussian Noise, Gaussian Blur and Random Crop. To make our watermark robust to those corruptions, we consider using all of these four corruptions as an augmentation in the training of each watermark detector. In

Table 5: Robustness of the watermark against different image corruptions

| Corruption | DreamBooth | | Textual Inversion | | Text-to-image | | LoRA | |
|---|---|---|---|---|---|---|---|---|
| Type | w/o aug. | w/ aug. | w/o aug. | w/ aug. | w/o aug. | w/ aug. | w/o aug. | w/ aug. |
| JPEG Comp. | 63.83% | 99.00% | 86.42% | 96.58% | 61.08% | 93.67% | 79.42% | 91.08% |
| Gaussian Noise | 68.50% | 99.25% | 90.17% | 97.75% | 91.08% | 91.67% | 75.83% | 86.17% |
| Gaussian Blur | 45.17% | 99.25% | 75.58% | 97.67% | 92.92% | 95.42% | 93.08% | 91.08% |
| Random Crop | 83.83% | 99.08% | 86.50% | 96.50% | 73.25% | 88.00% | 71.58% | 84.67% |

Table 6: Watermark detection rate (TPR) under different watermark rates

| WM Rate | DreamBooth | | | Textual Inversion | | | Text-to-image | | | LoRA | | |
|---|---|---|---|---|---|---|---|---|---|---|---|---|
| | Ours | GW | IM | Ours | GW | IM | Ours | GW | IM | Ours | GW | IM |
| 100 % | 99.66% | 98.47% | 91.5% | 96.54% | 87.78% | 88.28% | 98.74% | 93.33% | 88.89% | 97.49% | 85.83% | 87.83% |
| 80 % | 99.58% | 97.22% | 90.68% | 97.75% | 84.86% | 88.35% | 96.92% | 80.70% | 63.73% | 95.00% | 78.34% | 75.21% |
| 50 % | 95.92% | 94.45% | 54.91% | 92.92% | 83.20% | 37.60% | 86.25% | 77.50% | 48.75% | 73.75% | 66.11% | 64.39% |
| 20 % | 86.42% | 92.78% | 13.81% | 88.25% | 81.25% | 19.30% | 83.33% | 74.45% | 32.11% | 63.83% | 54.45% | 44.76% |

Table 5, we show the accuracy of the watermark detectors which are trained with or without augmentation on the corrupted images. From the table we can see that, the performance of the watermark detector on the corrupted images is substantially improved after the augmentation is applied in the training of the detector. After the augmentation, the classifier can achieve an accuracy near 100% against all the corruptions in DreamBooth's images. Even in the images generated by LoRA, where the classifier performs the worst, the accuracy can still be consistently higher than 84%.

## 4.6 ABLATION STUDIES

**Reduced watermark rate.** Watermark rate refers to the percentage of a dataset which is protected by the watermarks. In real practice, the data protector may have already released their unmarked images before the development of the watermark technique. Therefore, it is necessary to consider the situation where the watermark rate is not 100%. In this subsection, we demonstrate the effectiveness of FT-Shield when the watermark rate is lower than 100%. The experiments are mainly based on the style transfer task using the paintings by artist Louise Abbema. The results are demonstrated in Table 6. As shown by the results, as the percentage of the watermarked images in the training dataset decreases, the watermark detection rate in the generated images also decreases. This is within expectation because when there are less watermarked images in the protected dataset, it is harder for the watermark to be assimilated by the diffusion model. Nonetheless, our method consistently achieves watermark detection much higher than baselines. With a watermark rate of 80%, it can achieve a detection rate close to 100% across all fine-tuning methods. Even when the watermark rate is reduced to 25%, our method can still achieve detection rates higher than 80% across all the fine-tuning methods except LoRA. Although the watermark detection rate for LoRA's generated images experienced the most substantial decline, it remains much higher than the two baseline methods.

**Detector trained without data augmentation.** As discussed in 3.3, it is necessary to use the generated data from the fine-tuned diffusion model to augment the dataset used for the watermark detector's training. Table 7 demonstrates the performance of the classifier if there is no augmentation (based on the style transfer task). The classifier is simply trained on the dataset which contains the clean and watermarked protected images. According to the results demonstrated in Table 7, when there is no augmentation, the watermark detector can successfully detect some watermarked images on the generated set, especially those generated by DreamBooth. However, their performance is much worse than the ones with augmented data. This difference demonstrate the necessity to conduct augmentation when training the watermark detector.

Table 7: Performance of watermark detector trained without augmentation

| | | Ours (4/255) | Ours (2/255) |
|---|---|---|---|
| DreamBooth | TPR↑ | 84.67% | 79.06% |
| | FPR↓ | 9.56% | 9.61% |
| Textual Inversion | TPR↑ | 54.83% | 47.61% |
| | FPR↓ | 3.83% | 13.22% |
| Text-to-image | TPR↑ | 37.89% | 46.00% |
| | FPR↓ | 2.89% | 15.50% |
| LoRA | TPR↑ | 44.06% | 53.61% |
| | FPR↓ | 4.72% | 19.67% |

## 5 CONCLUSIONS

In this paper, we proposed a novel watermarking method to protect the intellectual property of images against the fine-tuning techniques of text-to-image diffusion models. Our method incorporates the training loss of the fine-tuning of diffusion models in the training objective of the watermarks. This ensures the early assimilation of watermarks by diffusion models, providing a more robust and reliable watermark framework in the protection of image copyright. Empirical results demonstrates the effectiveness of our method and its superiority over the existing watermarking methods.

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

# A ALGORITHM

The detailed algorithm for the training of the watermark (Equation 1) is demonstrated as below.

---

**Algorithm 1** Optimization for watermark $\delta_i$

---

**Input:** Protected dataset $\{x_i\}_{i \in [n]}$, Captions for protected dataset $\{c_i\}_{i \in [n]}$, Initialized watermark $\{\delta_{i,0}\}_{i \in [n]}$, Pretrained text-to-image diffusion model with parameters $\theta_1, \theta_2$ ($\theta_1$ denotes the unet part and $\theta_2$ denotes the other parts), watermark budget $\eta$, diffusion model learning rate $r$, batch size $bs$, PGD step $\alpha$ and epoch $E$

**Output:** Optimal watermark $\{\delta_i^*\}_{i \in [n]}$

1: **for** Epoch=1 to E **do**
2:     **for** Batch from $\{x_i\}_{i \in [n]}$ **do**
3:         $\theta_1^* \leftarrow \theta_1$
4:         **for** 1 to 5 **do**
5:             $\theta_1^* \leftarrow \theta_1^* - r \frac{\partial}{\partial \theta_1^*} L_{dm}\left(\theta_1^*, \theta_2, x_{1:bs}, c_{1:bs}\right)$       // Use clean images to update the unet
6:         **end for**
7:         **for** 1 to 5 **do**
8:             $\delta_{1:bs} \leftarrow \delta_{1:bs} - \alpha sign\{\frac{\partial}{\partial \delta_{1:bs}} L_{dm}\left(\theta_1^*, \theta_2, x_{1:bs} + \delta_{1:bs}, c_{1:bs}\right)\}$
9:             $\delta_{1:bs} \leftarrow Proj_{\|\delta_{1:bs}\|_\infty \leq \eta}(\delta_{1:bs})$      // PGD to update watermark
10:        **end for**
11:       **for** 1 to 5 **do**
12:         $\theta_1 \leftarrow \theta_1 - r \frac{\partial}{\partial \delta_{1:bs}} L_{dm}\left(\theta_1, \theta_2, x_{1:bs} + \delta_{1:bs}, c_{1:bs}\right)$ // Use watermarked images to update the unet
13:         **end for**
14:     **end for**
15: **end for**

---

# B ADDITIONAL DETAILS ABOUT THE FINE-TUNING METHODS

In the experiments of this paper, we considered four Fine-tuning methods of text-to-image models including DreamBooth, Textual Inversion, Text-to-Image and Text-to-Image-LoRA for style transfer and object transfer tasks. More details about the setting of these fine-tuning methods are provided as below.

- **DreamBooth (Ruiz et al., 2023)**: DreamBooth is a fine-tuning method to personalize text-to-image diffusion models. It mainly focus on fine-tuning the unet of the diffusion models' structure with a prior preservation loss to avoid overfitting and language-drift. Whether to update the text-encoder within in the text-to-image model structure is an open option. In the experiment in this paper, we update both the unet and the text-encoder with learning rate to be 2e-6, batch size to be 1 and maximum fine-tuning stpes to be 800. For style transfer task, we use "[V]" as the unique identifier for the specific style and incorporate "[V]" in the prompts in the sampling process to instruct the model to generated images following this style. Similarly, for object transfer, we use "sks" as the identifier.

- **Textual Inversion (Gal et al., 2022)**: Textual Inversion is another text-to-image diffusion model personalization method. It focuses on adding a new token which is connected to a specific style or object to the vocabulary of the text-to-image models. This work by using a few representative images to fine-tune the text embedding of the pipeline's text-encoder. In our experiment, we set the fine-tuning learning rate to be 5.0e-04, batch size to be 1 and maximum fine-tuning steps to be 1500. We use "[V]" as a placeholder to represent the new concepts that the fine-tuning process learn and also incorporate it in the prompts in the sampling process.

- **Text-to-Image**: Text-to-Image Fine-Tuning Method is a simple implementation of the fine-tuning of text-to-image diffusion models. It fine-tunes the whole unet structure with a dataset which contains both the images and the captions describing the contents of the images. An identifier "[V]" is also required in the captions in the fine-tuning and sampling procedure. In our experiment,

we set the learning rate for fine-tuning to be 5e-06, the batchsize to be 6 and the maximum fine-tuning steps to be 300.

- **LoRA (Hu et al., 2021)**: LoRA works by adding pairs of rank-decomposition matrices to existing weights of the UNet and only train the newly added weights in the fine-tuning process. It also required a image-caption pair dataset for fine-tuning and and the identifier "[V]" in the cations. In experiments, we set the learning rate to be 5e-06, batch size to be 6 and maximum training steps to be 3000.

## C   PROMPTS USED FOR IMAGES GENERATION

### C.1   PROMPTS USED FOR STYLE TRANSFER

| | |
|---|---|
| A lady reading on grass in the style of [V] | Anglers on the Seine River in the style of [V] |
| Flower field in the style of [V] | Haystacks in winter mornings in the style of [V] |
| Iris in the style of [V] | Mother and her child in a garden in the style of [V] |
| Red Boats at Argenteuil in the style of [V] | Saint Lazar Railway Station in the style of [V] |
| Sunflowers in the style of [V] | The Cliffs of Etelta in the style of [V] |
| Windmills on the Flower Field in the style of [V] | Boats at rest at petit gennevilliers in the style of [V] |
| Eese in the creek in the style of [V] | Meadow with poplars in the style of [V] |
| Claude haystack at giverny in the style of [V] | Olive tree wood in the moreno garden in the style of [V] |
| A fountain in the style of [V] | A bottle of champagne in an ice bucket in the style of [V] |
| Snow scene in the style of [V] | Sunrise in the style of [V] |
| The artist house in the style of [V] | The boat studio in the style of [V] |
| The bodmer oak fontainebleau in the style of [V] | The cabin in the style of [V] |
| Portrait of a Woman with Low Neckline in the style of [V] | The sea at saint adresse in the style of [V] |
| The seine in the style of [V] | The sheltered path in the style of [V] |
| The summer poppy field in the style of [V] | Walk in the meadows in the style of [V] |
| A dog waiting for the owner in the style of [V] | Waterloo bridge in the style of [V] |
| Woman in a garden in the style of [V] | Swans gliding on a serene pond in the style of [V] |
| Moonlit night over a calm sea in the style of [V] | Majestic castle overlooking a valley in the style of [V] |
| An old man playing the violin in the style of [V] | Wild horses galloping on the shore in the style of [V] |
| Cherry blossoms in full bloom in the style of [V] | Bustling train station in the 1900s in the style of [V] |
| An Italian vineyard at midday in the style of [V] | Birds taking flight from a tree in the style of [V] |
| Ballerinas rehearsing in the style of [V] | Market day in a provincial town in the style of [V] |
| The old lighthouse by the cliff in the style of [V] | A goat on grass in the style of [V] |
| Two butterflys in the style of [V] | A growling tiger in the style of [V] |
| A frog on a lotus Leaf in the style of [V] | A camel team in the desert in the style of [V] |
| An eggplant on vines in the style of [V] | Waterfall in the style of [V] |
| Alm tree at bordighera in the style of [V] | Rough sea in the style of [V] |
| A woman admiring lotus flowers in the style of [V] | Two pandas eating bamboos in the style of [V] |
| Grapes on a vine in the style of [V] | A view of mountain in the style of [V] |
| A man in a suit with a beard in the style of [V] | The side face of a red-haired woman in the style of [V] |

**Remarks**: In the sampling procedure of DreamBooth and Textual Inversion, the prompts need to be added with the word "painting", e.g., A lady reading on grass in the style of [V] painting.

### C.2   PROMPTS USED FOR FACE OBJECT TRANSFER

| | |
|---|---|
| A photo of [V] laughing heartily | A photo of [V] by the beach at sunset |
| A photo of [V] wearing A vintage hat | A photo of [V] with A glass of wine |
| A photo of [V] reading A thick book | A photo of [V] wearing A graduation cap |
| A photo of [V] with A colorful parrot on the shoulder | A photo of [V] holding A vintage camera |
| A photo of [V] holding A coffee cup | A photo of [V] on A boat |
| A photo of [V] holding A bottle of water | A photo of [V] wearing A winter scarf and gloves |
| A photo of [V] holding A bouquet of flowers | A photo of [V] wearing oversized sunglasses |
| A photo of [V] in A library, surrounded by towering bookshelves | A photo of [V] in the jungle |
| A photo of [V] surrounded by festive balloons | A photo of [V] amidst colorful autumn leaves |
| A photo of [V] in A sunny park | A photo of [V] in A classroom |
| A photo of [V] in her bedroom | A photo of [V] with A kitten |
| A photo of [V] with straight black hair | A photo of [V] with ear rings |
| A photo of [V] with blunt-cut bangs | A photo of [V] with upset face |
| A photo of [V] in the street | A photo of [V] in front of A window |
| A photo of [V] with short hair | A photo of [V] in front of A flower field |

### C.3 Prompts Used for lifeless object transfer

| | |
|---|---|
| A [V] in the snow | A [V] with a wheat field in the background |
| A [V] on the beach | A [V] with a tree and autumn leaves in the background |
| A [V] on a cobblestone street | A [V] with the Eiffel Tower in the background |
| A [V] on top of pink fabric | A [V] on top of green grass with sunflowers around it |
| A [V] on top of a wooden floor | A [V] on top of a mirror |
| A [V] with a city in the background | A [V] on top of a dirt road |
| A [V] with a mountain in the background | A [V] on top of a white rug |
| A [V] with a blue house in the background | A red [V] |
| A cube shaped [V] | A [V] placed beside a window |
| A girl holding a [V] | A [V] on a desk |
| A [V] on a chair | A [V] beside a computer |
| A [V] on the top of a roof | A [V] in a box |
| A [V] on a bed | A man with a [V] |
| A [V] on a desk | A [V] on a cliff overlooking the sea |
| A [V] placed on a bookshelf | A [V] under a tree |

