# OpenReview forum: "FT-SHIELD: A Watermark Against Unauthorized Fine-tuning in Text-to-Image Diffusion Models"
_ICLR.cc/2024/Conference — Submitted to ICLR 2024_

### Official Review · Reviewer_1g5B · 2023-10-29

**Soundness:** 3 good
**Presentation:** 2 fair
**Contribution:** 3 good
**Rating:** 3
**Confidence:** 4

**Summary:**

The authors introduce FT-Shield in this study, a sophisticated watermarking approach engineered to secure copyright adherence in text-to-image diffusion models against unauthorized fine-tuning. FT-Shield achieves this by inserting a watermark into images, which persists when adversaries employ these watermarked images for fine-tuning text-to-image models. The robustness of FT-Shield is validated across diverse fine-tuning scenarios, confirming its deterrent capability against unauthorized exploitation and its reinforcement of legal copyrights. This investigation signifies a notable progression in the protection of intellectual property within the field of generative modeling, contributing to responsible AI development.

**Strengths:**

1. This paper addresses a significant topic pertinent to the current landscape of generative modeling.
2. The authors endeavor to assess their methodology within realistic scenarios.
3. The manuscript is composed with clarity, offering a narrative that is both well-articulated and easy to follow.

**Weaknesses:**

1. While the authors' argument is intriguing, I would recommend expanding the experimental validation to more comprehensively substantiate the claims presented. Further details on the experimental design and outcomes would be particularly beneficial. (Please Refer questions)

2. The inclusion of additional qualitative results would greatly enhance the robustness of the study. In the current appendix, there is a limited variety of cases presented; for instance, Figure 3 showcases a singular style. Enriching this section with a broader array of cases, including those involving objects and more styles, would be advantageous.


Suggestions for Improvement:
1. To encapsulate a wider spectrum of applications, I would suggest incorporating tests on human images as well. Protecting human figures is important, and as such, it would be valuable to see examples, such as those involving public figures (e.g., Nicolas Cage).

**Questions:**

General questions
1. Is there a risk of the proposed watermark inadvertently manifesting in unrelated styles or objects? For instance, if style A is watermarked and then utilized by an adversary, there's a query whether a generated image with an unwatermarked style B could yield a false positive detection. Clarification on this possible form of FPR would be valuable.

2. Could the authors explore the feasibility of generating multiple watermarks within their framework? Given that practical applications often necessitate protecting a variety of styles or objects, understanding how the proposed system manages multiple watermark integrations is critical. Challenges such as the potential overwriting of previously learned watermarks or an increase in false positive rates (FPR) are of particular concern and merit discussion.


Questions for experiments
1. It is suggested that Table 3 includes FID scores to substantiate the authors' claim regarding adversaries potentially halting the fine-tuning process once personalization is achieved. Providing FID scores and corresponding visual results for each condition tested would offer a more complete analysis of the model's performance.

2. Could the authors specify what is meant by "one fine-tuning" as used in the context of Section 4.4 for assessing transferability? A more detailed explanation would help clarify the experimental procedures.

3. In Section 4.5, could you quantify the intensity of the disturbances applied and discuss how varying levels of disturbance strength influence the True Positive Rate (TPR)?

4.  in Section 4.5, would it be possible for the authors to present results where all disturbances are combined, to evaluate the cumulative effect on the watermark detection system?

---

### Official Review · Reviewer_WhCc · 2023-10-31

**Soundness:** 3 good
**Presentation:** 2 fair
**Contribution:** 2 fair
**Rating:** 6
**Confidence:** 3

**Summary:**

In this paper, the authors introduce a novel watermark designed to detect unauthorized fine-tuning in diffusion models. Unlike previous watermarks, the proposed watermark can be rapidly learned by diffusion models due to the integration of a diffusion objective within the optimization of watermarking perturbations. Simultaneously, the authors enhance the detector's robustness by training it with synthetic data and incorporating data augmentations.

**Strengths:**

- The issue addressed in this paper is very important.
- The design of the watermarking objective is intuitive, and the proposed method outperforms recent baselines significantly. I appreciate that the authors chose the most recent baselines for comparison, with one being a concurrent submission to ICLR 2024.
- The ablation studies are comprehensive.

**Weaknesses:**

- There is an absence of experiments on the transferability between various diffusion models. For instance, while the watermarks are trained with Stable Diffusion 1.5, they may be used with SDXL. It would be insightful to ascertain whether the watermark remains effective across different diffusion models or if a different model can learn it quickly.
- Minor suggestion: It might be advantageous to show the TPR at a fixed FPR, such as TPR@FPR=1%, in the tables that only present TPR. It would be easier to make comparisons between different methods or scenarios.

**Questions:**

- The experimental design seems somewhat ambiguous. The data protector's objective is to "detect if a suspected model is fine-tuned on the protected images." Yet, the entire experimental section centers on determining whether a given image was generated by a model fine-tuned on the protected images. Does this imply that to verify if a model has been fine-tuned with the protected images, the data protector simply generates one image from the target model and evaluates the detector's output? I believe a stronger detection could be achieved by generating multiple images during the detection phase.
- Would the watermark be robust under the attack from [1], where [1] uses a diffusion model to denoise the watermarking perturbation?

[1] Zhao, X., Zhang, K., Wang, Y. X., & Li, L. (2023). Generative Autoencoders as Watermark Attackers: Analyses of Vulnerabilities and Threats. arXiv preprint arXiv:2306.01953.

---

### Official Review · Reviewer_gF4L · 2023-10-31

**Soundness:** 2 fair
**Presentation:** 2 fair
**Contribution:** 2 fair
**Rating:** 3
**Confidence:** 5

**Summary:**

Diffusion models can be easily fine-tuned to achieve personalization. However, using unauthorized images as fine-tuning data can raise copyright concerns. In order to handle this issue, watermarking methods are proposed to identify whether a generated image is produced by a model fine-tuned with unauthorized data. Unluckily, when using existing watermarking methods, the style or object information is learned by the fine-tuned diffusion model earlier than the watermark. As a result, offenders can evade watermarks by reducing fine-tuning steps. In order to mitigate this issue, this paper proposes FT-Shield. In particular, FT-Shield optimizes the watermark such that it can be easily learned. Extensive experiments validate the performance of the proposed FT-Sheild. The idea of optimizing watermarks such that the watermarks are easier to learn is novel and interesting.
The proposed FT-Shield is verified to be effective for various fine-tuning methods, e.g., lora, dreambooth.

**Strengths:**

- The proposed FT-Shield is verified to be effective for various fine-tuning methods, e.g., lora, dreambooth.

**Weaknesses:**

- The idea of the paper does not seem novel.
- The authors claim that during fine-tuning, 1) existing watermarking methods learn style before watermark; 2) the proposed FT-Shield can learn watermark before style. However, no empirical evidence is shown.

**Questions:**

The images used to compute TPR and FPR are unclear. Could you please provide more related information, e.g., prompts and models used to produce images?

---

### Official Review · Reviewer_DTgk · 2023-11-02

**Soundness:** 2 fair
**Presentation:** 3 good
**Contribution:** 2 fair
**Rating:** 3
**Confidence:** 4

**Summary:**

This paper introduces a method for detecting
unauthorized fine-tuning in text-to-image diffusion models.
It involves embedding optimized watermarks into the images
being safeguarded, allowing the watermark patterns to be
quickly learned by the text-to-image diffusion models. A
trained binary classifier can then be used to identify the
unauthorized usages of the protected images. The
effectiveness of the proposed method was validated through
experiments on four fine-tuning techniques.

**Strengths:**

* The studied problem is intersting. Preventing unauthorized
diffusion model based image mimic is an important problem.

* The experiments incorporate various fine-tuning-based
image mimic methods, such as DreamBooth, LoRA, and Textual
Inversion.

**Weaknesses:**

* The novelty of this paper might be limited. The problem
settings and the proposed framework resemble those in
Gen-Watermark [1]. Except for the watermark creation
process, other elements like the overall workflow, decoder
construction and training, and using images generated by
fine-tuning methods to enhance decoder training, are similar
to Gen-Watermark [1]. The detailed approaches for generating
the watermarks in FT-Shield uses the bi-level optimization
(Equation 1) to make the watermarks can be learned faster.
However, this bi-level optimization method might be also
similar to the method proposed in DiffusionShield [2], which
also weakens the contribution of this paper.

* The bi-level optimization method described (Equation 1)
necessitates white-box access to the parameter of the
text-to-image diffusion model. Is the model utilized by the
protector the same as the one used by the offender in this
paper? Since the protector can only control the released
data and is unaware of the models the offender will employ,
it is suggested to consider conducting extensive evaluations
in scenarios where the models used by the protector and
offender differ significantly, such as in terms of model
sizes and architectures.

* The high performance shown in Table 1 presume that the
protector is aware of the fine-tuning method employed by the
offender and uses images generated by the corresponding
fine-tuning method to augment decoder training. However, the
accuracy drops substantially (from over 95% to approximately
75%) in more realistic settings where this assumption
doesn't apply, as seen in Table 4. Additionally, IM is not
assessed in Table 4. Given that most experiments are based
on this assumption, more extensive evaluation in more
realistic scenarios where this assumption does not hold is
suggested.

* Supporting data for Figure 1 is missing, and it remains
unclear how the completion of the style learning is defined.
Moreover, the offender might opt to fine-tune the model
adequately to fully mimic the style, in which case the
proposed method might perform similarly to existing
approaches. It is suggested to clearify the offender's
motivation for using fewer fine-tuning steps.

[1] Ma et al., Generative watermarking against unauthorized subject-driven image synthesis. arXiv 2023.

[2] Cui et al., DiffusionShield: A Watermark for Data Copyright Protection against Generative Diffusion Models. arXiv 2023.

**Questions:**

Please refer to the weaknesses.

---

### Meta-Review · Area_Chair_o2F7 · 2023-12-05

**Metareview:**

This paper introduces a method for detecting unauthorized fine-tuning in text-to-image diffusion models.

Strengths:
* The paper addresses the important concern of copyright and intellectual property in genAI image generation (DTgk,WhCc,1g5B)
* The method is demonstrated to work across different fine-tuning methods like LoRA, Dreambooth and textual inversion (DTgk,gF4L,WhCc)
* Ablation studies to validate the method (WhCc).

Weaknesses:
* The paper's approach closely resembles existing frameworks like Gen-Watermark (DTgk).
* The bi-level optimization method requires white-box access to the text2image diffusion model's parameters (DTgk).
* The high accuracy reported is contingent on the protector knowing the offender's fine-tuning method (DTgk)
* Lack of clarity (DTgk) and details on the method (1g5B)
* The paper does not include experiments to test transferability and the watermark's effectiveness across different diffusion models (gF4L,WhCc)
* Insufficient experiments on a variety of styles (1g5B) and no human valuation (1g5B)

Reviewers provided scores 3, 3, 3 and 6. The authors did not engage in the discussion nor provide a rebuttal. Based on these scores, I vote to reject the paper.

**Justification For Why Not Higher Score:**

Based on the scores (3, 3, 3, 6), the paper does not meet the bar for acceptance.

**Justification For Why Not Lower Score:**

N/A

---

### Decision · Program_Chairs · 2024-01-16

Reject